# A cost analysis of Machado-Joseph's disease (MJD)

**Cristiane da Silva**[1]*, **Marco Tulio Aniceto França**[2], **Giácomo Balbinotto Neto**[3]

**1** Polytechnic School, Undergraduate Academic Unit, University of Vale do Rio dos Sinos, São Leopoldo, Rio Grande do Sul, Brazil, **2** Department of Economics, Pontifical Catholic University of Rio Grande do Sul, Porto Alegre, Rio Grande do Sul, Brazil, **3** Department of Economics, Federal University of Rio Grande do Sul, Porto Alegre, Rio Grande do Sul, Brazil

☯ These authors contributed equally to this work.
* cristianesi@outlook.com

**Data Availability Statement:** All relevant data are within the paper and its Supporting information files.

**Funding:** This work was supported by the Coordenação de Aperfeiçoamento de Pessoal de Nível Superior - Brazil (CAPES) Code 001. The

## Abstract

A rare disease is that with a low prevalence in the population. However, it is estimated that there are between 6,000 and 8,000 different types of rare diseases in the world and, generally, they are incurable and deadly. Machado-Joseph's disease (MJD) is one of these cases; of genetic origin, autosomal dominant, with a high chance of transmission between generations and without curative treatment. Given the specificities of MJD and the lack of economic studies associated with it, this article aims to estimate the direct and indirect health-related costs of lost productivity attributable to Machado-Joseph's Disease. The data used were primarily collected at the Hospital de Clínicas de Porto Alegre (HCPA), during the period between October 2019 and March 2020. The *bottom-up* cost methodology was used, that is to say, to estimate costs across a sample of patients to produce an annual cost per patient. Among the main results, it was found that 90.8% of the sample does not work and of these, 72.73% reported that the reason they had stopped working was due to Machado-Joseph's disease. The average age of men when they stopped working was 39.05 years of age and for women it was 39.64 years of age. In relation to direct non-medical costs, with rehabilitation and transport, it was found that these items affect about 32% of men's income and 36% of women's income, as well as medication and diapers costs affect about 15% of women's income and 14% of the income of men who are no longer able to work. The study also showed that 50% of caregivers, who are generally close relatives of the patient, do not work. Of these, 33.3% reported having left the labor market to provide assistance to the patient, which means that the cost for families is even higher than that estimated for the patient.

## 1. Introduction

The analyzing of direct and indirect health costs for rare diseases are still incipient in Brazil. In this study, we seek to estimate the direct and indirect health-related costs of lost productivity attributable to Machado-Joseph's Disease. This is expected to contribute to the expansion of research that uses the economic approach, especially for a specific rare disease that, to our knowledge, has not received any attention in terms of economic evaluation.

funders had no role in study design, data collection and analysis, decision to publish, or preparation of the manuscript.

**Competing interests:** The authors have declared that no competing interests exist.

MJD is a rare, neurodegenerative, usually fatal and of late-onset disease [1]. Patients who are affected by the disease present difficulties in gait, posture, limb coordination, speech and sight-motor system [2]. The evolution of loss of movement ends up by binding them to a wheelchair. The progressive worsening of the clinical condition causes the loss of work capacity, so that many of them are forced to leave the job market early, even though, MJD does not affect the mental integrity of the bearers [3].

When it comes to MJD, one can think about the importance of studying its particular case, highlighting the scale of the health problem for society and the potential savings to families and society as a whole, if it could be eradicated, given its high prevalence in Rio Grande do Sul (RS) and the characteristic of it being an autosomal dominant disease. MJD has a prevalence of 6 affected per 100,000 inhabitants in Rio Grande do Sul [4].

In the literature, there is a greater concentration of studies on the cost of the disease for those that are chronic and prevalent. Numerous cost-of-disease studies have been conducted over the past three decades, but few have been specific to rare diseases. Although there is no universal definition for rare diseases and there is a consensus that such diseases result from their low prevalence in the population [5, 6].

The scarcity of studies makes it difficult to assess the true economic and socioeconomic burden of these diseases, which can serve as a reference and basis for the formulation and structuring of health policies and a regulatory framework for the sector. Studies of the cost of the disease and the economic measurement of the same on individuals and society contribute to envision the direct consequences of diseases for the health system and the indirect costs associated with the loss of productivity of the patient and the caregiver. In addition, these studies can be used as a tool to assist in prioritizing and justifying public health and prevention policies. The study of the cost of illness aims to determine the total economic impact (cost) of a disease or health condition on society, through the identification, measurement and evaluation of direct and indirect costs [5, 7–9].

The cost-of-illness analysis can indicate which interventions are most valuable in reducing or avoiding the economic burden, so that the distribution of public and private investments in health can be re-planned. Several sectors can benefit from disease cost studies, such as the government that may have elements to estimate the financial impact of a disease on the public budget for resource allocation, and pharmaceutical companies that can identify diseases with high costs and direct investments in research and development appropriately. In addition, the information is very useful for other economic evaluation studies such as cost-utility and cost-effectiveness. It should be noted that while cost-of-disease studies can identify and measure all the costs of a particular disease, they do not address issues of inefficiency or waste, as well as the cost-benefit ratio of interventions [5, 9].

Studying the economic cost of MJD makes it possible to measure the direct and indirect consequences of the disease not only for the patient, but for the family, the caregiver and society as a whole, depending on the perspective adopted. In addition, studies such as this one can provide subsidies for the formulation and structuring of public health and prevention policies aimed at reducing the economic burden generated by the disease and improving the quality of life of patients and their families. As an example, one can mention Bill 6500/19, presented on 12/17/2019 to the Board of Directors of the House of Representatives to amend Laws No. 7,713, of 1988; 8,036, 1990; 8,112, 1991; and 8,213, of 1991, and consider Machado-Joseph disease (autosomal dominant spino-cerebellar ataxia type 3) as a serious disease. The objective is to provide income tax exemption on income from retirement or retirement, movement of the worker's linked account in the FGTS, granting permanent disability retirement and sickness benefit regardless of deficiency period [10].

This article intends to contribute to this gap, in the sense of bringing primary data and encouraging future studies from an economic point of view for a disease as important as the case of MJD. We chose to use the bottom-up cost methodology, in which costs are estimated in a sample of patients to produce an annual cost per patient.

The data used are from patients with MJD treated at the Hospital de Clínicas de Porto Alegre (HCPA), residents of Rio Grande do Sul, a state with a high prevalence of the disease. This high prevalence is associated with a migratory process. The first description of the disease in 1972 occurred in an Azorean family and, later, in 1976, it was described as a new genetic entity also in a family of Azorean descent. The patients had similar symptoms, suggesting that they comprised a single genetic entity with variable phenotypes. The oldest traced lineage was from Antone Joseph, born in Portugal in 1815, who left many affected descendants in Northern California [11].

In addition to this introduction, this article will consist of three more sections. The first one discusses the empirical evidence of the cost analysis of rare diseases. In the second, the empirical strategy that will be carried out in the MJD cost analysis is presented. Then, the results and discussions are presented and, finally, in the last section, the final considerations related to the problem under study will be presented.

## 2. Empirical evidence: The costs of rare diseases

Rare diseases are defined as those that affect a small number of people. They are usually degenerative, chronically debilitating diseases and require ongoing treatment. The physical, mental, sensory and behavioural capabilities of the patient are often affected [12, 13]. The threshold of prevalence of such diseases varies in different countries and regions, with no single definition [12, 14]. In Brazil, the prevalence is defined as 65 people per 100,000 or 1.3 people per 2,000 [15].

In the case of rare diseases, it is common for patients to have late diagnosis and experiencing various barriers to access treatment, whether related to the authorization process to receive treatment, given the difficulty in finding enough patients to carry out clinical trials, direct, indirect and intangible costs involved, to the lack of regulation and funding for rare diseases. Furthermore, the cost of drugs is inversely proportional to the prevalence of the disease; in other words, the rarer the disease is, the higher the cost of treatment. There are some reasons for this, among them: a) the small number of patients who will use the drug; b) it is difficult to find enough patients for clinical trials [12, 16, 17].

The scarcity of economic studies related to rare diseases is not only a problem in Brazil, therefore, in this subsection, some empirical evidence related to the costs of rare diseases will be presented, as well as the implications of these for the patient and his family. The objective is to highlight the costs imposed by these diseases that fall not only on the patient, but on their family and society.

Table 1 presents a set of works: in the first column, the bibliographic reference consulted, in the second column what is being investigated in the article. In the third column, the costs reported in the sample investigated in each study and, in the last column, the respective results and implications of the study, the disease(s) and/or the procedure.

The studies point to the need to contribute to the formulation of public health policies that allow tracking the population at risk, guide them with genetic counselling and encourage the development of treatments for rare diseases.

In order to deepen the study of the costs of this type of disease, the next subsection deals specifically with Machado-Joseph Disease, a rare, degenerative disease with considerable probability of transmission to offspring and characteristics common to some other rare diseases,

**Table 1. Costs of rare diseases.**

| Author/s | What is being investigated | Costs/Perspective | Results and implications |
|---|---|---|---|
| [18] | The economic burden of 23 types of rare diseases. | The average annual cost for inpatients and outpatients was US$1,562.97 and US$166.19, respectively, for the period from January 2013 to December 2016. | The management of rare diseases presents a major challenge for health policy makers, physicians, patients and society in general due to treatment difficulties, gaps in knowledge, costs involved and access to medicines, among others. It seeks to contribute to research on rare diseases and raise awareness of their impact on society. |
| [16] | A rare disease called Mucopolissacaridose VI. | The cost per year per patient was estimated to range between €150,000 and €450,000 in 2008. | This disease has a genetic and hereditary origin. The disease implies severe disability and premature death. |
| [19] | The expansion of neonatal screening for newborns. | The total annual operating cost of screening 56,000 newborns for five specific diseases was estimated at €2.5 million or €45 per newborn when including the initial costs for building a new screening organization. Costs per quality-adjusted life year (QALY) earned is a maximum of €25,500. Preventing severe disability in a newborn would reduce costs to a maximum of €18,000 per QALY gained. | Among the benefits of neonatal screening, it is highlighted that early death could be prevented in one to three cases and severe disability in one to five cases. |
| [20] | Degenerative cerebellar ataxia (SCA). | The mean annual cost per patient with ACS was €18,776. | The disease has an effect on society in terms of mortality, morbidity and social aspects. |

Source: Prepared by the authors.

such as its origin being of a genetic and hereditary nature, the inexistence of medicine or treatment capable of cure, costs with transport for specialized follow-up, need for a caregiver, among others.

## 2.1 Analysis of costs of the Machado-Joseph Disease

Costs are calculated to estimate the resources or inputs that are used in the production of assets or services. However, it is important to be clear that the price charged to a payer is not necessarily the cost of the product or service. Thinking about a hospital system, the cost of treating a patient with a given diagnosis can have a considerable difference between the total cost to the hospital compared to the amount that the hospital charges the payer and compared to what is actually received from the payer. It is desirable that the amount reimbursed is higher than the cost borne by the hospital for providing the service, and lower than the standard amount charged that is set by the hospital [7].

Costs can be classified into four categories: direct medical costs, direct non-medical costs, indirect costs and intangible costs. Direct medical costs involve medical supplies that are used directly to provide treatment, such as pharmaceuticals, diagnostic tests, complementary tests, clinical care, care provided by pharmacists, care in the emergency department and hospitalizations, medications, treatment costs, relapse after patient discharge and management of complications in the medium and long term, salary of health professionals (doctors, pharmacists, nurses, etc.), rehabilitation, outpatient surgery, ambulance transport, among others [7–9, 21].

Direct non-medical costs are costs of patients and their families which are directly associated with treatment but are not medical in nature, such as the cost of transportation to and from the doctor's office, clinic or hospital, necessary food and accommodation, for patients and their families during treatments performed outside the city of residence, social services, informal assistance, payment of caregivers, investments in residences due to the illness suffered by the patient, hiring third parties to help with domestic tasks, among others.

Indirect costs involve costs that result from loss of productivity due to the illness or death, such as absence from work to receive treatment or lower productivity due to the effects of the

illness or its treatment. Intangible costs include costs of pain, suffering, anxiety or fatigue, fears, that arise from illness or treatment. It should be noted that, regarding this last category of costs, it is difficult to measure or assign a monetary value, which means that this type of cost is normally not considered in the analyzes given the difficulty of measurement and quantification [7–9, 21].

Before establishing which costs will be measured, it is necessary to determine the study perspective. Perspective is an economic term that describes who the relevant costs are based on the purpose of the study or in other words, who will pay for treatment. From the perspective of society, which is considered one of the most adequate and comprehensive, costs for the health insurance company, costs for the patient, costs of other sectors and indirect costs due to loss of productivity are included. If the perspective adopted is that of the payer, it may include costs for the health insurance provider or for the patient, or even a combination of both, when the patient has a co-participation in the costs of the health insurance provider [7, 8, 22].

The types of costs to be included in an economic evaluation will depend on the pathology and therapeutic options under evaluation, as well as the analysis perspective. If the perspective chosen is that of society as a whole, all types of costs (direct medical, direct non-medical and indirect) must be included regardless of who produces and finances them. However, if the chosen perspective is more restrictive, for example, the hospital, primary care management, etc., it will only be necessary to include the costs of interest to the entity [7, 8].

Cost studies can be carried out from different perspectives, such as hospitals, primary care centers, a specific hospital unit, a specific service within a hospital, among others. However, there are barriers to carrying out these studies in low and middle-income countries. The lack of financial conditions, incomplete records, lack of knowledge to carry out this type of study, are examples that can be cited. Therefore, the authors recommend applying a combination of *bottom-up* and micro costing approaches to cost items that comprise a large portion of total costs, suggesting creating data collection models, conducting interviews, and adapting existing data sources to perform analysis at the patient's level [23].

Three steps are carried out for the estimation and measurement of costs: a) identification of the most relevant costs to be incorporated in the evaluation; b) quantification of the volume of these resources, that is, at this stage the number of units consumed in the economic evaluation is estimated, the value that was consumed in natural units is recorded (number of days of hospitalization, number of visits to the emergency service, number and type of tests performed, etc.), and, c) monetary evaluation of resources consumed, since at this stage a cost (price) is attributed to the resources identified and quantified for each therapeutic option of the economic evaluation. When these prices are observable in the market and can be assumed to reflect the opportunity cost, it is sufficient to multiply the price of each resource by the number of times it was used to obtain the overall cost of the resource used. When these prices are not observable in the market, adjusted prices are used or quotations are used [8].

Another relevant point refers to cost adjustments in relation to time, which must be carried out when costs are estimated from information collected more than a year before the study [7]. This adjustment is also called cost standardization. Therefore, if retrospective data are used to evaluate resources employed over several years in the past, these costs must be adjusted or valued at a certain point in time. One of the ways of doing this is to consider unit costs at a certain point in time. Another way is to multiply all costs for the year the data was collected by the inflation rate for that year. When estimated costs are based on dollars spent or saved in future years, discounting is required. It should be noted that there is a time value associated with money. It is understood that people prefer to receive money today rather than sometime in the future. Thus, the money received today is more useful than the same amount received the next year [7, 8, 22].

There is a so-called time preference among people in a society. Even if there were no inflation and no other type of bank interest, people would prioritize living in the present as the future is uncertain and income is expected to increase. Therefore, when the costs of health assessment alternatives occur over a period of more than one year, it is necessary to convert them into monetary units equivalent to the year in which the analysis is carried out, the base year. Thus, the discount rate will assign less value to costs that will occur in the future [8]. That is, changes in this time value of money are estimated using a discount rate. It approximates the cost of capital given the interest rates on borrowed money. With this parameter it is possible to calculate the present value of expenses and future savings [7, 8].

The discount rates usually recommended for health care interventions are between 3% and 6%. When we vary these discount rates, a sensitivity analysis is performed [7]. It is important to highlight that the discount rate to be adopted in economic evaluations in health must reflect a social discount rate and not market interest rates. Social discount rates vary across countries and over time. As there is no general agreement on it, the rationality of the decision-making process and the economic environment of each society must be considered [24].

Six steps must be followed when estimating health costs, namely: 1) definition of the study perspective, that is, who will pay for the use of the technology or strategy in investigation; 2) delimitation of the time horizon, therefore, for how long the costs will be estimated; 3) identification of costs, therefore, which cost items will be included in the analysis; 4) measurement of costs, thus, what is the unit of measurement adopted for each cost item; 5) determination of the method for valuing costs, therefore, how values will be assigned to cost units; and, 6) temporary adjustments when the time horizon is greater than one year [25].

Recommendations for carrying out studies of disease costs in the Brazilian health system were proposed, addressing the types of costs, perspectives of analysis, presentation of monetary values and sources of information. The cost of illness method allows estimating the social impact of illnesses and injuries, combining direct costs (medical care, travel expenses, among others) and indirect costs (loss of productivity due to reduced working time, among others) in an overall estimate of the economic impact on society [21].

In relation to indirect costs, there are three types: a) those generated as a result of a reduction or absence of paid productivity due to illness (hours or days off work); b) those that occur due to a reduction in unpaid productivity due to illness, for example, due to a decrease in usual activities at home; c) those due to the loss of productivity of relatives and/or friends of the patient resulting from the need for care or monitoring of the patient [7–9].

When evaluating the losses in work productivity, it is necessary to consider absenteeism, which refers to the reduction of working hours within those provided for in a legal working day, and presenteeism, a situation in which the worker is present, in their workday, but with reduced productivity due to illness. This occurs when the person is working below the level of their abilities because they are not healthy. We emphasize the importance of valuing both presenteeism and absenteeism when quantifying the indirect costs of the disease [8].

The quantification of labor productivity loss can be performed using different instruments. Normaly, the number of days equivalent to lost workdays per month is used, in which productivity losses are calculated by the sum of absenteeism days, plus the "equivalent" in days of absenteeism, of the days on which the patient went to work. but with reduced productive capacity (presenteeism), by the following formula: $LWDE = W_1 + W_2 (1 - p)$, where $W_1$ refers to the number of days without being able to work due to illness (absenteeism), $W_2$ refers to the number of days worked with health problems (presenteeism), $p$ is the percentage of effectiveness at work and $(1 - p)$ the percentage of incapacity for work [8].

When it comes to the quantification of lost labor productivity, there are mainly two alternative methods: a) the human capital method in which indirect costs are quantified based on the

reduction in gross income of patients in the future due to morbidity or mortality produced by the disease and which are determined by the patient's salary; and, b) the friction cost method (or conjunctural costs) in which the amount of productivity lost as a result of illness depends on the period of time that companies need to find and train a replacement for the sick worker. In this second method, the more specialized the worker to be replaced, the greater the indirect cost [7–9, 22].

It is also important to differentiate financial costs from economic costs. Financial costs involve the use of real money to be used in resources that are needed to carry out a program or intervention, while the economic costs of an intervention includes, in addition to money, the value of resources for which no money was spent, the well-known opportunity cost, which is the use of the resource in its best alternative use [7–9, 21].

Opportunity costs are associated with opportunities left aside if resources are not used in the most profitable way [8, 9]. This cost from the perspective of the production cost that an individual could have in their work activity if they were not at home [22]. The resources committed to a product or service cannot be used with other products or services (opportunities) [7]. For example, one can think of a family member who provides assistance as an informal caregiver of a MJD patient, although no amount of money changes hands (the family caregiver is not paid), there is an opportunity cost associated with their care, because this family member could work, if he or she were not providing the assistance.

In studies involving the cost of diseases, they contribute to decisions in the allocation of health resources, since they provide information on the impact of the disease, allow the identification of research priorities, monitoring and evaluation, evaluate therapeutic differences (in cost-benefit analyses), help managers to properly analyze budgets and support the process of seeking efficiency in health systems [21].

## 3. Method

We use the *bottom-up* cost methodology to estimate costs in a sample of patients to produce an annual cost per patient. This choice is directly related to the information available for analysis. According to the methodological base for disease cost studies in Brazil, among the two existing ways to estimate each element of disease costs, the top-down method of measurement (*top-down*), which starts from the total values at the national level of the set of all diseases until arriving at the level where the cost of the disease under analysis is, generally the most convenient approach. However, given the particularity of the MJD, regarding the lack of information available in Datasus, the *bottom-up* approach was chosen [21].

To quantify the lost labor productivity in the case of MJD, we use the human capital method. This method has been applied to estimate the costs of lost paid work due to illness throughout an employee's lifetime until retirement. The method quantifies the impact of health care on lost work time, whether due to sick leave, presenteeism, employee morbidity or death. Loss due to absenteeism, morbidity and death are calculated as the employee's gross daily salary multiplied by the number of work days lost until retirement [7–9, 22].

However, a temporal adjustment of costs and results was made, since, in the economic evaluations, the time horizon of the study was longer than one year, as it considered the age at which the individual left the job market due to the MJD and the salary that stopped receiving in this period. This type of adjustment is necessary when the time horizon of the study is greater than one year so that future magnitudes are adjusted and expressed in their current value. When performed, this adjustment is made to both costs and health outcomes, to update them and transform them into present value, and consists of applying a discount rate to future values [7, 8, 22].

To analyze the direct and indirect costs attributable to Machado-Joseph Disease, data collection was organized in Microsoft Excel®, version 365, and statistical analysis was performed using a test for the difference between two means with independent samples assuming similar variances in the Stata®12 software. This version of the test is used when population deviations are not known but can be assumed to be similar. This version was preferred because it leads to a more powerful test, statistically [26]. In this test, the null hypothesis states that there is no difference between the means of the two populations in relation to each of the variables of interest, and the alternative hypothesis states that there is a difference between the means of the two populations in relation to each of the variables of interest, leading us to a two-tailed test.

### 3.1 Source of the data

Regarding Datasus as a data source, whenever consulting the parameters of the International Classification of Diseases (ICD 10), searching for vital statistics, general mortality, in the ICD-10 category, only the G11 category is found, corresponding to hereditary ataxia. But there is no information about the specific case of MJD, which deals with spino-cerebellar ataxia type 3 (SCA3). In addition, when the search is for health care, outpatient production, the procedure code 0301010196 refers to the clinical evaluation for the diagnosis of rare diseases axis 1—congenital anomalies, information that also does not specifically address MJD. Machado-Joseph Disease is not in the list of diseases in the Brazilian Unified Health System data sources.

Therefore, in order to obtain primary data, data obtained from patients and caregivers treated at the outpatient clinic of the Medical Genetics Service of the Hospital de Clínicas de Porto Alegre (HCPA) were used through the application of a structured questionnaire specially developed for this purpose. Currently, the HCPA concentrates information on patients with the disease and is a reference in research in the area. It is important to note that two pilot tests were carried out with the questionnaire prepared: i) with a family member affected by the disease and; ii) with a patient of the HCPA. Data collection took place between October 2019 and March 2020, and was carried out in person at the outpatient clinic before consultations and by telephone when necessary. In this study, there was interaction with caregivers and/or patients who answered the questions.

A structured questionnaire was used for the future realization of economic and statistical analyses, previously prepared, in partnership with the specialist physicians who care for patients with Machado-Joseph Disease. Some of the questions elaborated were based on the National Health Survey (PNS, 2013). After the patient's consent, a structured interview was carried out, focusing on economic and social issues of the individual and his/her caregiver (if any). For our study, we prepared the consent form for capable adults and the consent form for incapable adults. We also prepare the consent form for the use of image and voice. Consent was provided both in writing and verbally. When the patient was a minor, the responsible caregiver (father or mother) was responsible for him, with consent. Hospital de Clínicas de Porto Alegre has all these consents on file, as the hospital's medical genetics service constantly carries out research. The research was witnessed by Doctor Laura Jardim and her research team. This study was registered and submitted with CAAE 15740519.4.0000.5336 and CAAE 15740519.4.3001.5327, on Plataforma Brazil, to the Ethics Committee of the Pontifical Catholic University of Rio Grande do Sul (PUCRS) and Hospital de Clínicas de Porto Alegre (HCPA), respectively. In addition, it was also approved by the Scientific Committee of PUCRS and HCPA. After calculating the minimum sample size for this study, interviews were conducted with 109 out-patients of which 70 had a caregivers (64.22%), and two patients had two caregivers each. Most caregivers were women, 56.94% and 43.06% were men.

## 4. Results

The variables related to the evaluation of costs were as follows: for the evaluation of direct costs, the following were researched: expenditure on transport and food, medication and diapers, and follow-up appointments. In the analysis of indirect costs, the following were evaluated: the receipt or not of social benefit due to MJD (retirement or sick pay), the monthly amount and the loss of days worked by the patient. Intangible costs, although clearly existing in the case of MJD, were not considered in this study given the difficulty in assessing them, since they involve pain, suffering, fear of death, loss of expectations. Annual mean values were calculated for the entire group. The costs (in R$) are based on the year 2020.

Of the 109 patients with Machado-Joseph Disease, 61 (55.96%) were female and 48 (44.04%) were male. The mean age of women was 47.05 years and of men 45.25 years. According to the test of means, it was observed that there is no statistically significant difference regarding the mean age between genders. Analogous reasoning is performed for the other variables, so none of them reject the null hypothesis that there is equality between the female and male genders. It is also verified that the average number of family members with Machado-Joseph Disease was reported as 6.08 for women and 5.77 for men. The time elapsed since the onset of symptoms of the disease was calculated considering the difference between the variable "age at onset of symptoms" and the variable "current age of the patient", and the average observed for women was 10.75 years, with a standard deviation of 0.81 years and, for men, it was 9.73 years, with a standard deviation of 0.74 years. That is, on average, women in the sample have been living with the disease for approximately 11 years and men for approximately 10 years. As reported by patients, depending on their memory and degree of contact with family members, it was found that, on average, 6 individuals in the family had already received a diagnosis of MJD.

As for the individual income of MJD bearers who still work, which corresponds to 10 individuals out of the 109 affected, the vast majority (60%), for both genders, receive between R$ 1,050.00 and less than R$ 1,950.00. Among the patients who still work, only one of them has an informal professional activity. Among those who are out of the labor market and who receive some social assistance (such as disability retirement or some other), which corresponds to 86 individuals out of the total of 99 who do not work, and the rest do not receive any type of benefit nor do they work.

As for the thirteen patients who do not work and do not receive social benefits, as reported during the interviews, five of them reported working informally, without a formal contract, a 16-year-old girl never worked, she informed that she helped her mother in her work. The other seven patients reported not being able to access the social security benefit, either as sick pay or as disability retirement. It was found that 39.13% of women and 25% of men receive less than one minimum wage. In addition, the minority of the sample (10.47%), in general, receives more than R$ 2,850.00 per month. Data show that only 9.17% of those affected by MJD still remain in the labor market and they have an individual income higher than those who had to leave the labor market early, when compared to social security benefits and/or retirements.

By the *t* test for the difference between the two averages, it appears that the average income of women, who still work, corresponds to R$ 1,937.50 while that of men corresponds to R$ 2,154.00. It is important to note that a woman, a civil servant of the judiciary, was excluded from the sample for reporting a high income, constituting an *outlier* (R$ 12,000.00) influencing the average and distorting its result. It can also be seen that, on average, women left the labor market at 39.64 years of age and men at 39.05 years of age. Among those who receive some social security benefit, the amount received in the case of women is on average R$

1,800.75 and, in the case of men, it is R$ 1,405.83. In relation to those who receive a pension, among women the average is R$ 1,768.07 and for men it is R$ 1,839.76. Fig 1 presents the average costs associated with rehabilitation, transport and medication for men and women in general, also showing these costs separately, between those who work and those who do not. It is observed that the costs reported by women are, in general, higher than those reported by men.

The *t* test for the difference between means with independent samples assuming equivalent variances did not prove to be statistically significant for any of the analyzes performed. The p-value returned was always above the 5% significance level established in the study. In other words, it cannot be said that there is a statistically significant difference between the means of the two populations (women and men) in relation to each of the variables of interest (rehabilitation and transportation costs, medication costs—general, workers and non-workers). The exact p-value can be seen in Fig 1.

It is observed that the costs reported by women are, in general, higher than those reported by men. This may be due to the fact that women tend to take more care of their health compared to men. It is important to note that only the costs of those patients who reported having them were presented, that is, those who receive medication or diapers, transport and rehabilitation funded by the SUS, at no cost to the patient, were not considered here. In addition, it appears that the costs, especially with rehabilitation and transport, of those who still work are significantly lower than those of patients who are no longer able to work. This result may be associated with the difficulties imposed by the disease.

Although the analysis does not have scope for the most severe cases of MJD, since these patients no longer access the HCPA due to their worsening health conditions, there are indications that the severity of the disease is directly proportionate to its high costs, their Social

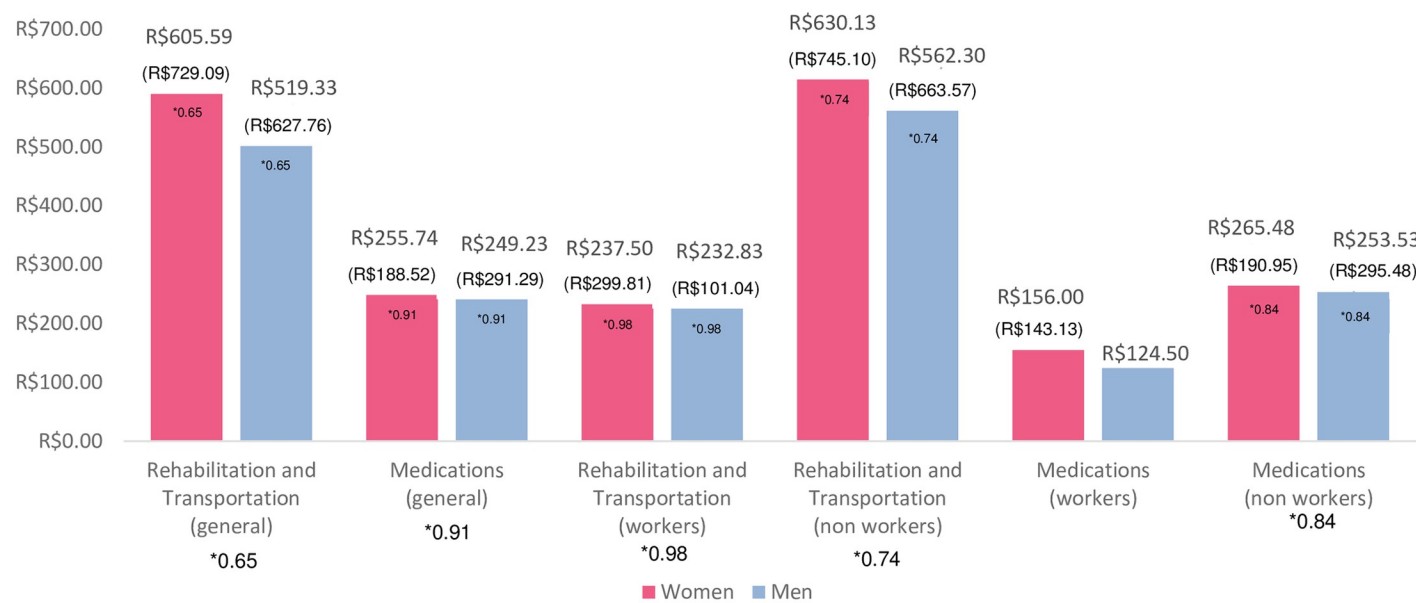

**Fig 1. Average costs with rehabilitation, transport and medication (by sex).** Note: 1. Standard deviation in parentheses. 2. * p value associate. Fig 1 highlights the costs generated by Machado Joseph's disease that fall on patients and their families. In terms of rehabilitation and transportation costs (general), the sample investigated was 55 patients. In terms of rehabilitation and transportation costs (workers), the sample investigated was 5 patients. In terms of rehabilitation and transportation costs (non-workers), the sample investigated was 50 patients. In medications costs (general), the sample investigated was 75 patients. In medication costs (workers), the sample investigated was 5 patients. It is worth noting that only one male patient who works reported having medication costs, which made it impossible to obtain estimates such as standard deviation and p value in the t test for two samples assuming equivalent variances. In medication costs (non-workers), the sample investigated was 70 patients. Source: Elaborated by the authors.

Security or Retirement Benefit. An analysis was performed stratifying the average cost by time of onset of symptoms, since the cost would tend to be higher the longer the time elapsed. However, as there is no scope for more severe cases of the disease (bedridden, for example), no differences in mean costs were observed for 3, 5, 10 and 11 years or more from the onset of symptoms. For patients in which the time of onset of MJD symptoms was less than or equal to three years, the direct cost was R$ 339.90 and the indirect cost was R$ 1,948.75. From four to five years after the onset of symptoms, the direct cost was R$ 411.50 and the indirect cost was R$ 1,759.46. From six to ten years from the onset of symptoms, the direct cost was R$ 364.43 and the indirect cost was R$ 1,741.65. And, for those aged 11 years or over from the onset of symptoms, the direct cost was R$ 466.49 and the indirect cost was R$ 1,802.00. Among men who still work, the cost of rehabilitation and transport is approximately 11% of their monthly income. And, among those who do not work, the expense represents approximately 32% of their social security or retirement benefit.

As for palliative medication expenses, for women who are still working, this expense represents approximately 8% of their monthly income and, among those who do not work, approximately 15% of their social security or retirement benefits. Among men who still work, the cost of medication represents approximately 6% of their monthly income and approximately 14% of their social security or retirement benefits for those who do not work.

In relation to direct costs, with rehabilitation and transport, and with medicines and diapers, it was found that the average monthly cost of each patient corresponds to R$ 605.59 and R$ 255.74, for women, and R$ 519.33 and R$ 249.23, respectively for men. It is observed that the expenditure on rehabilitation and transport is more than double the expenditure on medicines and diapers. This result can be associated with easier access to medication, through programs such as the popular pharmacy, for example, than physical therapists, speech therapists, psychologists and other specialized professionals.

When compared to the average monthly amount received as a result of social benefits, the significant weight that these costs have on the patients' budget and consequently on their family nucleus can be seen. Rehabilitation and transportation costs affect about 32% of the income of men and 36% of the income of women who are no longer able to work, and the costs of medicines and diapers affect about 15% of the income of women and 14% of the income of men who are no longer able to work, costs that should increase with the progression of the disease. It can be seen that for women, the commitment of income to expenses involving rehabilitation, transport and medication is higher than that of men.

Regarding marital status, it was observed that 57.38% of women and 47.92% of men are married. Only five of the married women (14.3%) and two married men (8.7%) do not have any children. Considering that this is a disease of genetic and hereditary origin, in which there is a 65% chance of transmission to the offspring, according to Souza et al. (2016) [4]; it is noteworthy that 81 individuals, representing 74.3% of the sample, have one to three children. With a high chance of intergenerational transmission and disoriented pregnancies, it is possible that the direct and indirect costs of MJD will continue to increase over the years. As for schooling, 56.25% of men have less than complete high school education, against 44.26% of women. Women are also ahead in terms of finishing high school and higher education when compared to the opposite sex.

When observing how many and what percentage of patients with MJD who reported direct and indirect costs associated with the disease, it appears that 55 of the patients who undergo rehabilitation claim to have some cost. Among those who undergo rehabilitation, 23 of them report having no associated cost, as they receive support through the city hall of the city in which they live. Regarding expenses with medicines and diapers, 31.19% of patients reported having no cost, while 68.81% reported costs ranging from less than R$ 50.00 to R$ 1,999.00.

Here, a debate becomes important: when a patient cannot afford the costs of a treatment, or at least part of it, the hospital of the public health system will bear the costs that are, in most cases, passed on in the form of higher costs for health plan insurers and for other patients, resulting in a socialization of losses.

Of the 109 patients in the sample, 70 of them have a caregiver (64.22%), and two patients have two caregivers each. Most caregivers are female, 56.94% and 43.06% are male. Regarding the degree of kinship, 51.4% of caregivers are spouses, 13.9% are children, 11.1% are mothers, 2.8% are fathers and 11.1% are other relatives of the patients. Only 6.42% of patients have a paid caregiver, with an average salary of R$ 1,007.86. Only one of the paid caregivers has a degree of kinship with the patient, being the patient's sister-in-law. All paid caregivers are informal, that is, they are workers without a formal contract. It was found that approximately 66% of patients already have a caregiver. The mean age of these patients is 46.3 years, in an age range that varies from 16 years to 79 years. Another very important result is that 33% of caregivers reported having left the job market to help the patient.

It is important to mention that the provision of informal care can have negative and positive effects on the caregiver's well-being. On the one hand, providing care can be costly, as caregivers invest time with the patient, sometimes they can perform tasks they are not comfortable with, they can develop physical and mental health problems, have financial difficulties, etc. On the other hand, caregivers may enjoy taking care of their family members, in addition, it generates an economic impact on society, as the time spent on providing informal care has an opportunity cost. In other words, from a social perspective, informal care translates into cost savings for society [27].

Patients who still work represent 9.2% of the total sample. Among those who do not work, 83.72% receive a pension and 16.28% receive some social security benefit. In addition, 11.93% of all patients in the sample are unemployed and have no income. In this case, those patients who are not working and who also do not receive any benefit were taken into account.

In the case of MJD, a disabling disease that generates a loss of productivity, it was observed that 90.8% of the sample does not work and of these, 72.73% report that the reason why they stopped working was due to Machado-Joseph's Disease. It is important to note that 89% of those who are currently no longer in the work market, were working before the symptoms appeared. In addition, the mean age of patients in the sample is 45.25 years for men and 47.05 years for women. Men have been out of the job market for approximately 6 years, and women for approximately 7 years.

When taking into account that the current age of retirement in Brazil, according to the Ministry of Economy (2020), is 65 for men and 60 for women and, that 80.81% of bearers of MJD stopped working before the age of 50, it becomes relevant to evaluate the age at which the individuals quit work.

The data showed that, among those who reported leaving work because of MJD, the average age of men when they stopped working was 39.05 years and, for women, it was 39.64 years, that is, there is an average loss of approximately 26 years of work for men and 20 years for women, without considering that many people continue to work after retirement, which in the case of MJD is practically impossible. This is because the disease is chronically debilitating, making it difficult to perform basic activities of daily living and especially working life.

Considering the average salary of R$ 1,981.82, received in Porto Alegre in 2021, by people who exercise the same professions reported by patients, or, correlated to them, these patients would be no longer receiving approximately R$ 24,866.32 per year. This calculation considers the inflation calculated by the 12-month accumulated Extended Consumer Price Index (IPCA), which, in January 2021, corresponded to 4.56%. Assuming constant inflation over the next few years, women would not receive approximately R$ 497,326.49 and, men, R$

646,524.43, in all these lost years of work. Applying a discount rate of 5%, women would not receive approximately R$ 196,808.98 and, men, R$ 190,920.46 in all these lost years of work.

In Table 2, the total direct and indirect costs and per patient are estimated for the year 2020. As the cost occurs in the current year, discount rates were not applied as suggested by the literature when the time horizon of the study is greater than one year [7–9, 22, 24]. These costs are annual and are in Reais. The estimate of indirect costs took into account sick leave and retirement. It should be noted that the national minimum wage for this period was R$ 1,045.00.

Table 2 reports a total annual cost per patient of R$ 9,871.90 for rehabilitation, transportation, medication and diapers. Per month, each patient spent an average of R$ 569.52 on rehabilitation and transport and, on average, R$ 253.14 on medication and/or diapers. Another important result is the total annual indirect costs due to retirement and social security benefits, which altogether reach an amount of R$ 1,830,954.00 in social terms.

As for the cost related to the patient's loss of work days for consultation, only the 9 patients who were in the job market on the date of the interview were analyzed, and this was R$ 82.31 per patient (obs: one patient was excluded from the sample ($n = 10$) for having an extremely high income and influencing the results of this group–civil servant of the judiciary). The average daily workload of these patients was 8 hours; five of them worked, at the time of the interview, in the private sector, two in the public sector, one was an entrepreneur and another, a worker without a formal contract.

Table 2 reports total annual direct and indirect costs, as well as costs per patient. In terms of rehabilitation and transportation costs, the sample included 55 patients who reported having this expense, while medication costs were reported by 75 patients. Regarding indirect costs, 72 patients receive a retirement, and 14 patients receive another social security benefit. Regarding the cost related to the patient's loss of working days for consultation, only the 9 patients who were in the job market on the date of the interview were analyzed.

**Table 2. Direct and indirect costs (total and per patient) of MJD patients.**

| | Total cost for 2020 (per patient) | Percentage of total costs (2020) |
|---|---|---|
| **Direct annual costs (R$)** | | |
| Rehabilitation and transportation | 375,882.00 (6,834.22) | 62.26% (69.23%) |
| Medication and/or diapers | 227,826.00 (3,037.68) | 37.74% (30.77%) |
| Total | 603,708.00 (9,871.90) | |
| **Indirect Annual Costs (R$)** | | |
| Retirement | 1556862.00 (21,623.08) | 85% (52.38%) |
| Social security benefit | 274,092.00 (19,578.00) | 14.96% (47.42%) |
| Lost work day (due to Consultation) | 740.80 (82.31) | 0.04% (0.20%) |
| Total | 1831694.80 (41,283.39) | |

Source: Elaborated by the authors.

Values expressed in Reais

* Parentheses indicate the total cost per patient.

Among the professions practiced are: entrepreneur, teacher, doorman, school assistant, cleaning assistant, analyst and housekeeper on a farm. When comparing the hourly rate of each of the professionals in the sample with the hourly wage of the respective professions (or similar to them when not located in CAGED) in RS, calculated according to the average working hours, by positions and professions, registered in the General Register of Employed and Unemployed (CAGED), it was found that approximately 56% of the sample receives an hourly wage lower than the average of their respective city in the state of RS.

It is important to mention that only one working day was lost, as this is the frequency of attendance at the HCPA genetics service, one annual consultation per patient. However, it should be noted that this cost can be higher, since patients usually undergo rehabilitation, follow-up with psychologists, speech therapists, physiotherapists, ophthalmologists, among others, on other days, in addition to this annual consultation. So, the cost of these professionals was underestimated. Another important point to consider with regard to cost estimation is the fact that the interviews were conducted with patients who still manage to go to the HCPA, that is, they are not undergoing the most severe phase of the disease; therefore, they are not bedridden. This is an indication that there will be more costs in the future.

According to the data collected, 44.04% of the patients with MJD in the sample use some instrument or support/aid to walk. Of these, 31.25% use a walker, 39.58% use a cane, 25% crutches, 2.08% use another means of support for walking and 2.08% use a walker and a cane. In addition, 38.53% of those interviewed already use a wheelchair. Some patients reported not using any instrument or support to walk because they had not adapted to them and, that they usually lean on the walls and furniture of their house. The costs of these instruments and/or support to walk were estimated through the SUS Procedures, Medicines and OPM (Orthoses, Prostheses and Special Materials) Management System (SIGTAP). It was found that if patients purchased instruments and/or supports, the unit cost for those using a walker would be R$ 130.00, the cost for those using a cane would be R$ 79.95, the cost for those using crutches, would be R$ 79.95 and for those using a wheelchair it would be R$ 571.90. The total cost to society of all patients using a walker, would be R$ 1,950.00; the cost of those using a cane, would be R$ 1,519.05; the cost of those using a crutch would be R$ 959.40 and, for those who use a wheelchair, it would be R$ 24,019.80, generating a total cost of R$ 28,448.25 if the purchase were made through the SUS. Another aspect that deserves to be highlighted is, that some of those affected with MJD, reported not using a wheelchair in order to avoid the beginning of its use, as much as possible. Patients report that, due to family experience, from the moment they need to use a wheelchair, they will not be able to avoid it, although they are aware that this will occur in the near future.

With regard to mobility issues, a study showed that patients with physical disabilities have their locomotion hindered or prevented due to the disease and that approximately R$ 175,000.00 in investment would be needed to meet demands for assistive technology equipment. It is also noteworthy that the SUS only finances the purchase of so-called standard wheelchairs; however, in some cases, adaptations are necessary to fully fulfill the function, meeting the specific needs of each patient [28].

The environment in which the individual interacts can be a facilitator or become a barrier for him to be able to carry out his activities with autonomy and independence. In other words, in addition to the disease itself, the lack of an adequate environment can generate insecurity, loss of autonomy and offer risks to the patient [29].

Among the necessary adaptations for patients with ataxias, the following can be mentioned: i) **entrances and doorways**: construction of ramps, handrails, corridors and wide doorways, handles that can be opened with the elbow; ii) **general circulation** around the house: non-slippery, uneven or multi-level floors; well-lit areas, including at night; sensors in corridors that

light up with movement; placement of colored bands on stairs, if any; furniture with rounded corners or use of corner protectors; bars attached to the wall for support; bells, tinklers or telephones with large numbers to call for help in case of falls; iii) **the bathroom**: bars in the shower and next to the toilet; non-slip floors; if necessary, raising the height of the W.C. by means of adapted seats or the construction of a masonry base; taps adapted to guarantee autonomy; bath chair with backrest; cordless phone or a help-call bell; bars attached to the wall; iv) **the kitchen**: shelves and appliances at shoulder height; long-handled grasper for picking up light items overhead; substitution of glass utensils by unbreakable materials [29].

The authors emphasize that these are ideal recommendations, but that financial constraints or the characteristics of the disease can often make their implementation difficult. The objective is always to make the environment safe, offering a certain degree of independence to patients, so that they can participate socially in the home environment, since a good life quality is not disconnected from feeling comfortable in their own home [29].

Instrumental activities of daily living (IADLs) that involve preparing meals, sweeping the floor, dusting the furniture, planning and organizing family and friends get-togethers, etc., require adaptations in the environment so that people with ataxia have more security and independence, even contributing to reduce the burden of a caregiver (and/or family member), in addition to preventing falls [29].

The performance of individuals with ataxia is affected in all daily activities, from the simpler ones, such as drinking a glass of water, to the work environment. Many of these individuals find it difficult to enter the job market due to the need to adapt the tasks performed according to the remaining abilities of the person with ataxia, while others need to temporarily or permanently withdraw from their work activities due to the symptoms of the disease. Adaptations similar to those discussed for activities of daily living at home extend to the work environment, which may further discourage hiring people with ataxia [29].

Such needs reinforce the importance of structuring public health policies to meet this demand and provide autonomy and independence to patients with ataxias, as is the case with Machado-Joseph Disease. The results of the statistical analyzes carried out in this subsection allowed us to observe that approximately 9.17% of patients with MJD are still in the job market. Considering the importance that work activity has in people's lives, it seems relevant to think about alternatives to support workers with MJD in the activities they develop in the work environment, as well as to reflect on adaptations to new functions as the limitations of the disease are perceived, respecting the skills and worker's interests.

The costs observed with rehabilitation, transport and medication were higher for women and there are indications that they increase with the severity of the disease. This hypothesis seems to be confirmed by finding that among those who are still able to work, the costs were lower than among those who are no longer able to work. In addition, with the progression of the disease's severity, mobility becomes more restricted, with the need to have a caregiver and to provide adaptations in the home environment.

The studies mentioned above are essential from a socioeconomic point of view, not only for individuals with the disease mentioned in each case and for their families, but also for society and for managers of the public health system who work with limited resources and need to be aware of disease-attributable costs to budget efficiently and responsibly. It is important to mention that, in the case of MJD, it was found that 50% of caregivers, who, in general, are close family members of the patient, do not work. Of these, 33.3% reported having left the job market to provide assistance to the sick, which means that the cost to families is even higher than what we have mentioned so far.

In addition, the information collected is from patients who still attend annual consultations at the Hospital de Clínicas de Porto Alegre, that is, the costs may be underestimated since we

did not have access to the most severe cases of the disease, such as bedridden patients, for example. In a longitudinal analysis, knowing patients' CAG repeat length is important, as it is a significant risk factor for faster progression in SCA3 [30].

This information is available in the patients' medical records, but given the context of the pandemic that took place at the time of preparation and in order to obtain the information for this article, it was not possible to access the data. In any case, it will not be an aggravating factor for the results, since this study is *cross-section*; therefore, the data are cross-sectional at a given point in time. Ergo, it does not consider the stages of disease evolution.

The *bottom-up* cost analysis approach used in this study is limited by the fact that it does not start from the national total values of the set of all diseases until reaching the level of the disease under analysis, which would be a more complete and convenient approach. On the other hand, considering the particularities of MJD, related to its rarity, the data available for analysis and the sample investigated, the methodology chosen can be considered satisfactory, efficient and robust, generating reliable data.

Another limitation of the study refers to the lack of data on direct medical costs resulting from outpatient and hospital care, exams, counselling and consultations, nursing services, among others, as Datasus cannot supply this information specifically for MJD. And it was not possible to obtain these data from the HCPA to carry out the analysis. This is left as a suggestion for future research.

## 5. Final considerations

The purpose of this article was to estimate the direct and indirect health-related costs due to lost productivity attributable to MJD. It started with an empirical review related to the cost analysis of rare diseases, to then develop the unprecedented cost analysis for Machado-Joseph Disease. One of the results found was that 70.51% of patients undergoing rehabilitation, and requiring transportation, reported some associated cost. While 68.81% reported having costs with medicines and diapers and 6.42% with caregiver costs. For the year 2020, in which the research was in progress, the total annual direct costs per patient were R$ 9,871.90. The total annual indirect costs with pensions and social security benefits amounted to R$ 1,830,954.00 from the public coffers.

The variable referring to the labor market shows that most of the interviewees no longer hold a professional activity and receive some social assistance (such as disability retirement or other assistance). However, it was identified that the income of both genders is less than a minimum wage and that only 10.47% of respondents earn more than R$ 2,850.00 per month. As reported by patients, depending on their memory and degree of contact with family members, it was found that, on average, 6 individuals in the family had already been diagnosed with MJD.

Another important analysis reported that 74.3% of the sample has one to three children, which reinforces the argument that actions are needed with the families regarding family planning, since the probability of transmission of MJD to children is high. The high genetic burden of transmission causes the costs of the disease to be perpetuated, compromising many generations in the same family. A characteristic that draws attention in the sample is that 95.41% of interviewees reported having at least one case of MJD in the family.

Costs were estimated using a specially developed questionnaire to deal with MJD, since in the Brazilian Unified Health System (SUS) data sources there is no information available specifically for this type of ataxia. Therefore, it is suggested that the Ministry of Health should include this rare disease, in a disaggregated way, in its SUS information systems.

This study had several points that deserve to be highlighted here, in addition to its originality, as it works with a significant sample of people affected by MJD. It is important to highlight that not only patients but also their families and society itself suffer from the high costs of MJD. Thus, the present study sought to estimate the direct and indirect costs due to lost productivity attributable to the MJD for the years 2019–2020. Both in the Brazilian and international cases, research involving the study of MJD costs is incipient, although in the medical context this disease is widely discussed and studied.

However, it is worth mentioning some limitations. The sample of patients analyzed includes only those who have or have had annual follow-up at the HCPA, analyzing data from a single referral center. And, it depends on the interviewees' memory regarding the monthly costs they have with the disease, the age they left the job market as well as the age at which the first symptoms appeared and how many people in the family have already had the disease. A cross-section analysis was carried out in a short period (2019–2020), therefore, it does not consider the stages of disease evolution that would allow a more in-depth analysis, by disease severity. Unfortunately, data on direct medical costs were not available, which would allow for greater detailing of MJD costs.

Nonetheless, this article contributes to the design and structuring of specific policies, such as the need for greater support for families in terms of rehabilitation and transport, specialized care closer to their homes, granting medicines and diapers in a less bureaucratic way, accessible to all who need them. In this regard, it is worth mentioning that some patients reported difficulties due to bureaucratic issues to obtain diapers, for example, and when they do, the quantity is not enough, so this cost ends up falling on the families.

Another relevant implication of this study lies in the purpose of promoting discussions and reflections for the elaboration of public health policies that make SUS-assisted reproduction accessible to patients, in order to interrupt the intergenerational transmission of the disease. The family needs constant guidance, support and backing so that they can make their decisions according to the possibilities that exist to avoid transmission between generations that cause pain, suffering and high costs associated with MJD.

Rare diseases, although they have received more attention over the last few years, suffer from the lack of public policies and social projects aimed at their needs. Although Machado-Joseph disease is one of these rare diseases and there is no cure or treatment capable of stagnating it, several services could be expanded, such as the intensification and dissemination of information about the causes and consequences of the disease, as well as the high rate of intergenerational transmission, offering rehabilitation and more accessible transport to patients so that they do not have to travel long distances, offering palliative drugs and diapers for those who need them without so much bureaucracy, planning for the care that will be needed in the mid term, with the need for caregivers reflecting on the impact on the family budget.

In Brazil, Bill 6500/2019, pending in the Chamber of Deputies, comes as an important instrument to guarantee benefits such as sick pay, disability retirement, income tax exemption and withdrawal from the Severance Indemnity Fund., to those affected with Machado-Joseph Disease. The change in the laws with the inclusion of MJD as a serious disease will allow access to benefits that until now (before it is sanctioned) were granted to a small specific group of diseases such as multiple sclerosis, irreversible and disabling paralysis, Parkinson's disease, among others.

At the time of writing this article, the bill status is listed as awaiting designation of rapporteur in the work, administration and public service committee. If sanctioned, it could ensure that individuals with MJD have their rights respected, allowing them to access benefits that are currently not available to everyone. In the investigated sample, approximately 13% of patients who are no longer able to work also do not receive any type of social benefit.

Finally, it is suggested that further studies consider an economic analysis of MJD to verify the imposed needs, the associated costs, the consequences for the family, patient, society and government.

## Supporting information

**S1 File.**
(XLSX)

## Author Contributions

**Conceptualization:** Cristiane da Silva.

**Data curation:** Cristiane da Silva, Marco Tulio Aniceto França.

**Formal analysis:** Cristiane da Silva, Marco Tulio Aniceto França, Giácomo Balbinotto Neto.

**Funding acquisition:** Marco Tulio Aniceto França, Giácomo Balbinotto Neto.

**Investigation:** Cristiane da Silva.

**Methodology:** Cristiane da Silva, Marco Tulio Aniceto França, Giácomo Balbinotto Neto.

**Project administration:** Cristiane da Silva, Giácomo Balbinotto Neto.

**Resources:** Cristiane da Silva.

**Software:** Marco Tulio Aniceto França.

**Supervision:** Marco Tulio Aniceto França, Giácomo Balbinotto Neto.

**Validation:** Marco Tulio Aniceto França, Giácomo Balbinotto Neto.

**Visualization:** Marco Tulio Aniceto França.

**Writing – original draft:** Cristiane da Silva.

**Writing – review & editing:** Cristiane da Silva.

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
