## [Decision Letter · Decision Letter 0]

7 May 2024

PONE-D-23-29643A COST ANALYSIS OF MACHADO-JOSEPH'S DISEASE (MJD)PLOS ONE

Dear Dr. Silva,

Thank you for submitting your manuscript to PLOS ONE. After careful consideration, we feel that it has merit but does not fully meet PLOS ONE’s publication criteria as it currently stands. Therefore, we invite you to submit a revised version of the manuscript that addresses the points raised during the review process.

Thank you for your article submission to PlosONE. We apologize for the delay in our response. Due to a reviewer becoming unresponsive, we needed to reassess the manuscript. Your article has been reviewed by three individuals; however, except for reviewer 1, the others have opted to remain anonymous for the study.

We look forward to receiving your revised manuscript.

Kind regards,

Asif Ali

Academic Editor

PLOS ONE

Journal requirements

Additional Editor Comments:

Reviewers' comments:

Reviewer's Responses to Questions

**Comments to the Author**

1. Is the manuscript technically sound, and do the data support the conclusions?

Reviewer #1: Yes

Reviewer #2: Partly

Reviewer #3: Yes

2. Has the statistical analysis been performed appropriately and rigorously? 

Reviewer #1: I Don't Know

Reviewer #2: No

Reviewer #3: Yes

3. Have the authors made all data underlying the findings in their manuscript fully available?

Reviewer #1: Yes

Reviewer #2: Yes

Reviewer #3: Yes

4. Is the manuscript presented in an intelligible fashion and written in standard English?

Reviewer #1: Yes

Reviewer #2: Yes

Reviewer #3: Yes

5. Review Comments to the Author

Reviewer #1: I don't believe this to be a rigorous research endeavor. No valuable insights are provided. I am not sure if this rudimentary economic analysis of MACHADO-JOSEPH'S DISEASE (MJD) makes it fit for the publication.

Reviewer #2: The authors intent to calculate the direct and indirect health-related expenses linked to lost productivity caused by Machado-Joseph’s Disease (MJD). This study may serve to amplify voices urging authorities to provide increased supportive care for families affected by rare diseases such as MJD. The authors must revise the manuscript in accordance with the identified revision points below:

1. Please avoid any citation in the abstract section, therefore, please move citation 1 to the introduction, if necessary

2. "Please incorporate more formal language into your phrasing, for example, avoid phrases like 'one can think that' or 'which is good news for the hospital'.

3. Please indicate how many caregivers and in- or out-patients are involved in the statistics.

4. In your statistical analysis, you should add the exact p value you found.

5. The first paragraph of the results section should move to the material and methods, and avoid to use future tense in the materials and methods.

6. Indicate your standard deviation or standard error as well as p value in the Graph 1 and explain it properly in the figure legend.

7. All figure legends should be moved beneath their respective tables or graphs. Each legend must include a concise explanation of the figure, including details such as statistical analysis and sample size.

Reviewer #3: The manuscript " A COST ANALYSIS OF MACHADO-JOSEPH'S DISEASE" by Silva et al. is a good analysis about MACHADO-JOSEPH'S DISEASE from cost perspective. This manuscript is of interest to the clinical studies, human biology researchers, as well as different biology researchers and I expect that the article will be well-cited. I have only one comment to consider. Authors add method and statistical details in their figures and tables legend and statistical section in materials and methods.

6. PLOS authors have the option to publish the peer review history of their article (what does this mean?). If published, this will include your full peer review and any attached files.

Reviewer #1: No

Reviewer #2: **Yes: **Sera Averbek

Reviewer #3: **Yes: **Pawan Kumar

---

## [Author Response · Author response to Decision Letter 0]

22 Jun 2024

Dear Plos One Editor,

We are grateful for the comments provided by the reviewers. Each of them has been carefully analyzed and considered relevant to improve the text. We appreciate the effort and contribution of each point raised. 

Thus, we are sending you the manuscript entitled "A cost analysis of Machado-Joseph’s disease (MJD)" that was written by me, with co-author Marco Túlio Aniceto França e Giácomo Balbinotto Neto, with the adjustments requested by the reviewers. The following is a list of the comments sent to us and our response to them:

Please confirm at this time whether or not your submission contains all raw data required to replicate the results of your study. Authors must share the “minimal data set” for their submission.

Response: In this review, we forward the database, with the information necessary to replicate all study results, as well as the related metadata and methods. In Excel, in addition to the raw data, there is the t-Test for averages, where the mean, variance, standard deviation, among other statistics, can be observed. We also make cost analyzes and projections available, as well as graphs and tables for the general public and by gender.

Reviewer #1: I don't believe this to be a rigorous research endeavor. No valuable insights are provided. I am not sure if this rudimentary economic analysis of MACHADO-JOSEPH'S DISEASE (MJD) makes it fit for the publication.

Response: This is one of the few works, perhaps the only one, that deals with the costs of Machado Joseph's illness in economic terms. The original contribution lies in the fact that it is one of the few analyzes carried out in the world and in Latin America. The analysis, although rudimentary, since the data is rudimentary, was carried out using information collected rigorously and undergoing evaluations by two ethics committees. Furthermore, this study may serve to amplify voices urging authorities to provide increased supportive care for families affected by rare diseases such as MJD.

Reviewer #2: The authors intent to calculate the direct and indirect health-related expenses linked to lost productivity caused by Machado-Joseph’s Disease (MJD). This study may serve to amplify voices urging authorities to provide increased supportive care for families affected by rare diseases such as MJD. The authors must revise the manuscript in accordance with the identified revision points below:

1. Please avoid any citation in the abstract section, therefore, please move citation 1 to the introduction, if necessary.

Response: We relocated the citation that was in the abstract to the introduction.

2. "Please incorporate more formal language into your phrasing, for example, avoid phrases like 'one can think that' or 'which is good news for the hospital'.

Response: We reviewed the entire text and tried to incorporate more formal language, as advised by the reviewer.

3. Please indicate how many caregivers and in- or out-patients are involved in the statistics.

Response: We complement the last paragraph of subsection 3.1 "Source of the data", as follows: “After calculating the minimum sample size for this study, interviews were conducted with 109 out-patients of which 70 had a caregiver (64.22%), and two patients had two caregivers each. 

Most caregivers were women, 56.94% and 43.06% were men.".

4. In your statistical analysis, you should add the exact p value you found.

Response: We have added the exact p value found in Fig 1, in the results section. Additionally, we included a paragraph just below Fig 1, explaining the result of the t-test and the respective p-values found.

5. The first paragraph of the results section should move to the material and methods, and avoid to use future tense in the materials and methods. 

Response: We moved the first paragraph of the results section to the last paragraph of the materials and methods section. We have revised the entire materials and methods section to avoid the use of future tense.

6. Indicate your standard deviation or standard error as well as p value in the Graph 1 and explain it properly in the figure legend.

Response: We indicate the standard deviation, as well as the p value in Figure 1 and correctly explain each item in the caption. We replaced the expression Graph 1 with the expression Figure 1.

7. All figure legends should be moved beneath their respective tables or graphs. Each legend must include a concise explanation of the figure, including details such as statistical analysis and sample size.

Response: We moved all captions so that they are below their respective tables or figures. We include in each caption a concise explanation of the figure and table, with details about the statistical analysis and sample size.

Reviewer #3: The manuscript " A COST ANALYSIS OF MACHADO-JOSEPH'S DISEASE" by Silva et al. is a good analysis about MACHADO-JOSEPH'S DISEASE from cost perspective. This manuscript is of interest to the clinical studies, human biology researchers, as well as different biology researchers and I expect that the article will be well-cited. I have only one comment to consider. Authors add method and statistical details in their figures and tables legend and statistical section in materials and methods.

Response: We indicate the standard deviation, as well as the p value in Figure 1 and correctly explain each item in the caption. We replaced the expression Graph 1 with the expression Figure 1. We include in each caption a concise explanation of the figure and table, with details about the statistical analysis and sample size. Additionally, we included a paragraph just below Fig 1, explaining the result of the t-test and the respective p-values found.

Sincerely, 

Cristiane da Silva

---

## [Decision Letter · Decision Letter 1]

15 Jul 2024

A cost analysis of Machado-Joseph's disease (MJD)

PONE-D-23-29643R1

Dear Dr. Silva,

We’re pleased to inform you that your manuscript has been judged scientifically suitable for publication and will be formally accepted for publication once it meets all outstanding technical requirements.

Kind regards,

Asif Ali

Academic Editor

PLOS ONE

Additional Editor Comments (optional):

Reviewers' comments:

Reviewer's Responses to Questions

**Comments to the Author**

1. If the authors have adequately addressed your comments raised in a previous round of review and you feel that this manuscript is now acceptable for publication, you may indicate that here to bypass the “Comments to the Author” section, enter your conflict of interest statement in the “Confidential to Editor” section, and submit your "Accept" recommendation.

Reviewer #2: All comments have been addressed

2. Is the manuscript technically sound, and do the data support the conclusions?

Reviewer #2: Yes

3. Has the statistical analysis been performed appropriately and rigorously? 

Reviewer #2: Yes

4. Have the authors made all data underlying the findings in their manuscript fully available?

Reviewer #2: Yes

5. Is the manuscript presented in an intelligible fashion and written in standard English?

Reviewer #2: Yes

6. Review Comments to the Author

Reviewer #2: (No Response)

7. PLOS authors have the option to publish the peer review history of their article (what does this mean?). If published, this will include your full peer review and any attached files.

Reviewer #2: **Yes: **Sera Averbek

---

## [Editor Report · Acceptance letter]

24 Jul 2024

PONE-D-23-29643R1 

PLOS ONE

Dear Dr. Silva, 

I'm pleased to inform you that your manuscript has been deemed suitable for publication in PLOS ONE. Congratulations! Your manuscript is now being handed over to our production team.

Kind regards, 

on behalf of

Dr. Asif Ali 

Academic Editor

PLOS ONE